# The Role of Hepatitis Viruses as Drivers of Hepatocancerogenesis

**DOI:** 10.3390/cancers16081505

**Published:** 2024-04-15

**Authors:** Mario Capasso, Valentina Cossiga, Maria Guarino, Luisa Ranieri, Filomena Morisco

**Affiliations:** Diseases of the Liver and Biliary System Unit, Department of Clinical Medicine and Surgery, University of Naples Federico II, 80131 Naples, Italy; mario.capasso2@unina.it (M.C.); maria.guarino@unina.it (M.G.); luis.ranieri@studenti.unina.it (L.R.); filomena.morisco@unina.it (F.M.)

**Keywords:** HCC, HCV, HBV, liver carcinogenesis

## Abstract

**Simple Summary:**

Despite the increasing incidence of HCC related to metabolism-associated steatotic liver disease (MASLD), viral hepatitis remains the major driver in liver carcinogenesis. Both HCV and HBV viruses have direct oncogenic properties that promote carcinogenesis, especially in active infections. The aim of this review is to summarize the viral mechanisms involved in liver cancer and to evaluate the changing incidence of HCC after antiviral treatment.

**Abstract:**

Recently, metabolic associated steatotic liver disease (MASLD) became the leading cause of chronic liver disease worldwide and one of the most frequent causes of hepatocellular carcinoma (HCC). Nonetheless, in this epidemiological trend, viral hepatitis remains the major driver in hepatic carcinogenesis. Globally, hepatitis B virus (HBV) is the leading cause of hepatocellular carcinoma, with an overall attributable risk of approximately 40%, followed by hepatitis C virus (HCV), which accounts for 28–30% of cases, with significant geographic variations between the Eastern and Western world. Considering all the etiologies, HCC risk increases proportionally with the progression of liver disease, but the risk is consistently higher in patients with viral triggers. This evidence indicates that both direct (due to the oncogenic properties of the viruses) and indirect (through the mechanisms of chronic inflammation that lead to cirrhosis) mechanisms are involved, alongside the presence of co-factors contributing to liver damage (smoking, alcohol, and metabolic factors) that synergistically enhance the oncogenic process. The aim of this review is to analyze the oncogenic role of hepatitis viruses in the liver, evaluating epidemiological changes and direct and indirect viral mechanisms that lead to liver cancer.

## 1. Introduction

### 1.1. Epidemiology

Hepatocellular carcinoma (HCC) is the seventh most common tumor based on incidence worldwide. Fortunately, its incidence seems to have lowered with an overall average percentage decrease of −1.93% compared to 1990s [1]. This trend is likely linked to vaccination (for HBV) and DAA-based treatment (for HCV) [2]. However, it still represents the third leading cause of cancer-related mortality worldwide (8.3% of all cases) [3,4].

It is well documented that globally HBV is the primary cause of HCC; it accounts for the highest incidence of liver cancer cases and fatalities worldwide (33%), followed by alcohol (30%), HCV (21%), and other causes (16%), with substantial geographic variations. Notably, the attributable risk for HBV is 60% in Africa and East Asia, whereas it is 20% in the Western world, where HCV infection is identified as the most common underlying liver disease etiology, with its prevalence ranging from 29% to 44% [5]. In Italy, data from ITA.LI.CA study group [6] demonstrate a progressive increase in non-viral cases, in accordance with the global epidemiological trend. Nevertheless, viral hepatitis remains responsible for over two-thirds of HCC cases [7]. In the last decade, HBV accounted for only 8% of the diagnosed HCC cases, while HCV was implicated in approximately 43% (Figure 1). Although dysmetabolism plays an increasingly relevant role in hepatocarcinogenesis, hepatotropic viruses remain the major drivers in the epidemiological landscape of HCC. All the most recent guidelines agree that cirrhosis is an independent and the highest-risk condition for HCC occurrence; however, the risk is higher in patients with virus-related liver disease [8].

### 1.2. Non-Viral Co-Factors

In this scenario, non-viral co-factors can also influence the HCC risk. In patients with viral infection, co-factors could accelerate the occurrence of HCC. Particularly, drinking alcohol, smoking, and a dysmetabolic phenotype are the main non-viral risk factors for HCC. As reported in Figure 1B, in Italy, these co-factors are responsible for almost 10% of the adjunctive risk of HCC in patients with viral infection.

Alcohol consumption increases the risk of HCC, proportionally with daily intake. Indeed, heavy drinking significantly intensified the hepatocarcinogenesis process through constant liver damage promoted by NF-kB activity [9]. The odds ratio (OR) for HCC occurrence in patients that are heavy drinkers (more than 60 g/day) and have HBV is significantly higher than in patients without relevant alcohol intake (48.6, 95% CI: 24.1–98 vs. 22.8, 95% CI: 12.1–42.8) [10]. In patients with HCV, heavy alcohol consumption increases OR to 126 (95% CI: 42.8–379), while it is 26.1 (95% CI: 12.6–54) in patients with a lower alcohol intake (0–40 g/day) [11]. Moreover, a higher 5-year cumulative incidence rate of HCC (30%, *p* = 0.0009) was observed in HCV-related cirrhotic patients with active alcohol consumption without SVR compared to patients with SVR [12].

Smoking seems to have a similar synergistic effect on the HCC risk. Tobacco may increase liver toxicity since it contains numerous carcinogens, including cadmium, whose relationship with HCC is well known [9]. Related to viral hepatitis, a more than multiplicative interaction with the HCC risk was found for HCV infection and cigarettes. The ORs based on the HCV status and smoking were HCV-/ever-smokers 1.50 (95% CI: 1.25–1.80), HCV+/never-smokers 7.94 (95% CI: 4.40–14.3), and HCV+/ever-smokers 23.1 (95% CI: 9.43–56.8), with a multiplicative interaction index of 1.60. Furthermore, a more than additive interaction with the HCC risk was demonstrated for HBsAg positivity and smoking (ORs: HBsAg−/ever-smokers 1.87 (95% CI: 1.30–2.69), HBsAg+/never-smokers 15.8 (95% CI: 9.69–25.7), and HBsAg+/ever-smokers 21.6 (95% CI: 15.2–30.5), with an additive interaction index of 1.44 [13,14].

In the era of increasing rates of diabetes and obesity, it becomes essential to delineate their role in HCC development.

In a large Taiwanese cohort of patients with HBV retrospectively followed-up to evaluate the HCC risk, a significant difference was found in the HCC incidence between patients with more than three metabolic risk factors and patient with fewer metabolic factors (13.6% vs. 4.83%, *p* = 0.003); furthermore, insulin resistance had the largest effect on the HCC risk in these patients [15].

Glycemic dysregulation is the main metabolic factor responsible for this hepatocarcinogenesis process. Indeed, insulin resistance could induce the pro-inflammatory adipokine and oncogene pathway, leading to inflammation, fibrogenesis, and hepatocarcinogenesis, through the well-known mechanisms underlying the pathogenesis of HCC-NASH [16,17]. In the general population, diabetes has been significantly associated with a higher cumulative incidence of HCC, and it could be considered as an independent risk factor, particularly when diabetic complications and insulin resistance are present [16]. Two metanalyses showed that diabetes was associated with a risk ratio of 2–2.3 for HCC development [18,19]. Moreover, in a large multiethnic cohort study conducted in 257.649 patients with diabetes and 772.947 patients without diabetes, the HCC incidence was significantly higher in patients with diabetes (2.39 per 10.000 person-years in patients with diabetes vs. 0.87 per 10.000 person-years in patients without diabetes, *p* < 0.0001). Diabetes was also associated with a greater than two-fold increase in the relative risk of HCC in the multivariate proportional hazards analysis [20]. Recent systematic reviews and meta-analyses further demonstrated the key role of diabetes as a metabolic comorbidity that raised the HCC risk in virus-infected patients. In these patients, the risk increased further because diabetes worsened liver fibrosis, and it seemed to be slightly higher in patients with HCV than in patients with HBV [17]. Indeed, the accurate meta-analysis by Chao Y. et al. strongly underlined the role of diabetes on HCC development, showing that in patients with HBV the cumulative risk for HCC was an RR of 1.37 (95% CI: 1.24–1.51) in cohort studies and an OR of 1.99 (95% CI: 0.73–5.48) in case–control studies, while in patients with HCV, it was an RR of 1.76 (95% CI: 1.42–2.17) in cohort studies and an OR of 1.77 (95% CI: 1.18 to 2.64) in case–control studies [21]. Many other studies confirm these results: in a meta-analysis conducted on 21.842 patients (five cohort studies and two case–control studies) identified a pooled hazard ratio (HR) of 1.77 (95% CI: 1.28–2.47) in patients with HBV [22] and a systematic review of seven studies in 10.700 patients with HCV reported that diabetes increased the risk of HCC by two-fold (effect sizes ranging from HR = 1.73; 95% CI: 1.30–2.30, to relative risk (RR) = 3.52; 95% CI: 1.29–9.24) [23].

Being overweight/obesity is also associated with the HCC risk. Oxidative hepatic environment in obesity models has been associated with an increased functioning of the pro-inflammatory pathway, promoting hepatic fibrosis and has been correlated with HCC development [24]. A recent investigation involving Korean adults with HBV found that obesity was linked to a 22% higher HCC risk in men (HR: 1.22; 95% CI: 1.09–1.36) and to a 46% higher risk in women (HR: 1.46; 95% CI: 1.24–1.71) compared to patients without obesity [25]. In an American cohort of 865 patients with HCV–cirrhosis, the adjusted HR (aHR) for HCC development was 3.4 (95% CI: 1.5–7.5) for patients who were overweight and 2.2 (95% CI, 0.9–5.0) for patients who were obese [26]. These results were similar to those obtained in a French cohort of 220 patients with HCV and cirrhosis, in which a significant positive linear relationship between BMI levels and the HCC risk was observed. Particularly, a BMI of 25–30 kg/m^2^ (HR: 1.7; 95% CI: 1.0–2.8; *p*: 0.049) and BMI of 30 kg/m^2^ or more (HR: 2.9; 95% CI: 1.6–5.5; *p*: 0.001) were risk factors for HCC development [27].

To summarize, lifestyle plays a key role in hepatocarcinogenesis. It is imperative for individuals to avoid alcohol consumption and smoking, as well as to focus on weight management and proper diabetes management. These measures are essential for reducing liver damage, particularly in patients with viral-related liver disease. By prioritizing these interventions, individuals can potentially reduce their HCC risk and improve their overall liver health. Therefore, comprehensive lifestyle adjustments should be emphasized as integral components of disease management strategies for these kinds of patients.

### 1.3. Viral Pathogenesis

The mechanisms of HBV and HCV hepatocarcinogenesis are detailed in the following sections. Anyway, independent of the etiology of liver disease, to appropriately define the HCC risk, a liver status evaluation should be performed. Indeed, the presence of compensated cirrhosis at presentation and the sustained activity of viruses are significant predictors of HCC in viral-related cirrhosis. Fibrosis and the consequent cirrhosis are both expressions of the virus-related indirect oncogenic mechanisms. However, there is a non-negligible percentage of patients who develop HCC without underlying cirrhosis: approximately 20% of HCC occurs in a non-cirrhotic liver. In virus-related diseases, HCC can develop in a non-cirrhotic liver due to direct oncogenic mechanisms. In fact, the incidence of HBV-related HCC in patients without cirrhosis is only 10% lower than in patients with cirrhosis, while the incidence of HCC in patients without cirrhosis and with HCV is 30% lower [28]. Furthermore, co-infections such as HBV/HCV and HBV/HDV increase the risk of HCC (by two- to six-fold relative to each infection) [29].

## 2. Hepatitis C Virus

### 2.1. Mechanisms of Hepatocarcinogenesis

HCV belongs to the Flaviviridae RNA virus family. It includes six major genotypes (1–6), having a genetic diversity from 31 to 35% [30]. HCV-related hepatocarcinogenesis is a long process, generally resulting from decades of chronic infection through the subsequent chronic inflammation and fibrosis [31]. The presence of underlying cirrhosis in almost all cases of HCV-related HCC led to the hypothesis that HCV induced carcinogenesis mostly through indirect mechanisms, specifically inflammation linked to cirrhosis. Nowadays, several studies demonstrated HCV as an independent risk factor for HCC, even if the exact mechanisms are not yet completely understood [29]. The role of immune alterations has been highlighted. Specific HCV proteins could induce the dysregulation of cell surveillance, the induction of stem-like cells, and alterations in apoptosis signaling as potential HCC drivers [32,33].

During an HCV infection, the immune-mediated liver injury significantly contributes to carcinogenesis, inducing the spread of cancer-related mutations and the expansion of abnormal cells. Moreover, viral proteins stimulate oncogene expression, cell cycle dysregulation, and the deactivation of tumor suppressor genes, and this results in the proliferation of quiescent hepatocytes [34,35,36]. Furthermore, the failure to eliminate HCV-infected cells leads to viral variants that could induce an immune escape mechanism, facilitating cancer development [37].

Lastly, unlike HBV, HCV exerts its oncogenic activity through its structural proteins that have been implicated in HCC pathogenesis. Particularly, the core protein has the ability to modulate intracellular pathways (such as the activation of nuclear factor kappa B pathway) and to upregulate several cellular proteins (such as interleukin-IL-6, signal transducer and activator of transcription-STAT-3), whose dysregulation induces transformational changes in hepatocytes [38,39]. Some studies also suggested a minor role of non-structural HCV proteins by inhibiting apoptosis and increasing hepatocyte proliferation [34,40].

### 2.2. Incidence and Risk Factors of HCC in Patients with HCV

HCC is the prevalent complication and the first cause of death in patients with HCV-related cirrhosis, while a chronic HCV infection is the most common underlying liver disease among patients with HCC worldwide (mostly in North America, Europe, and Japan) [41]. In patients with HCV, HCC develops almost always in the presence of cirrhosis. Japanese studies demonstrated that the HCC risk was four-fold higher in patients with cirrhosis (7.1 per 100 person-years) than in subjects without cirrhosis (1.8 per 100 person-years) [42,43]. In Europe and the United States, the incidence rate of HCC is 3.7 per 100 patient-years in patients with HCV-related cirrhosis, but it is not possible to estimate the rate in patients with chronic hepatitis because of the lack of HCC cases in this group [29].

As already shown, HCV has oncogenic effects, which result in a greater HCC risk compared to other liver disease etiologies. In a large American cohort of patients with different etiologies of cirrhosis, Ioannou et al. [44] showed that patients with HCV had a more than three-fold increased HCC incidence (3.3 per 100 patient-years) than patients with alcoholic (0.86 per 100 patient-years) or metabolic cirrhosis (0.90 per 100 patient-years).

### 2.3. HCC Risk in Patients with SVR

The advent of direct-acting antivirals (DAAs) dramatically changed HCV history, with the achievement of sustained virological response (SVR) in the majority (>95%) of patients treated for HCV [45]. Several studies demonstrated that the incidence of HCC was significantly lower in patients with SVR. Particularly, in a large cohort of 2249 patients with HCV-related cirrhosis that were treated with DAAs, Calvaruso V. et al. showed that the rate of HCC occurrence was higher in patients without SVR than in patient with SVR (12.8% vs. 3%, *p* < 0.001) [46]. Similar results were obtained in an American cohort with HCV treated with DAAs, underling that the accomplishment of SVR significantly reduced the HCC risk (0.90 vs. 3.45 HCC/100 person-years; adjusted HR: 0.28, 95% CI: 0.22–0.36) [47]. However, the newest DAA-based antiviral treatments, aimed to eradicate HCV, reduce but do not eliminate the HCC risk, especially in patients with cirrhosis or advanced fibrosis. Long-term follow-up in patients with cirrhosis and an SVR showed a 2–5 times lower risk of HCC. In a study by El Serag et al. involving over 10,000 American Veterans with SVR, the annual risk of HCC in patients with cirrhosis was 1.39% and remained constant over time [48]. In European cohorts of patients with HCV-related cirrhosis, the incidence rate of HCC after SVR was between 1.6 and 2.3 person-years, confirming a residual and not negligible HCC risk [49,50].

Several studies tried to individualize factors associated with a higher risk of developing HCC after SVR because not all patients with cirrhosis exhibited the same risk. In a prospective study on 687 cirrhotic patients achieving SVR, the sole independent predictor of HCC was a baseline liver stiffness measurement (LSM) of >20 kPa [49]. Many studies also demonstrated that albumin levels of <3.5 g/dL and a platelet count of <120.000 are associated with an increased HCC risk, both as pre-treatment and post-treatment variables [46,50]. Moreover, the presence of additional comorbidities that impact the HCC risk (e.g., diabetes, obesity, and alcohol use) maintains the risk at a higher level compared to patients without comorbidities [51].

Finally, a very recent study showed that the HCC risk declined progressively up to 6-years after SVR. Indeed, in patients with cirrhosis, the HCC incidence was 2.71 per 100 person-years in subjects accruing 1–2 years after SVR, while it was 1.65 per 100 person-year in patients accruing >4 to 6 years after SVR. Among subjects without cirrhosis, HCC risk did not have a significant association with time since SVR [52].

Table 1 presents the most recent and accurate studies that evaluated HCC incidence in patients with HCV.

In conclusion, achieving SVR in patients with HCV-related cirrhosis is associated with a decreased HCC risk over time. Despite this reduction, the residual HCC risk remains elevated and surpasses the thresholds deemed necessary for continued surveillance and screening. These results underscore the ongoing importance of monitoring patients with SVR and HCV in order to detect and manage any potential development of HCC in a timely manner. Therefore, continuing vigilance and adherence to screening protocols even after SVR achievement in this patient population is mandatory, particularly in patients with co-factors or with high pre-treatment stigmata.

## 3. Hepatitis B Virus

### 3.1. Mechanisms of Hepatocarcinogenesis

Hepatitis B virus belongs to Hepadnaviridae family, and its structure is like a double-strained DNA virus. The molecular mechanism underlying HBV-related hepatocarcinogenesis is still intricate and involves genetic and epigenetic changes in the host DNA, the inhibition of repair mechanisms, and the promotion of cell proliferation by altering cellular signaling pathways. After infection, HBV converts its DNA into a covalently closed circular DNA (cccDNA), which accumulates in the nucleus of hepatocytes as a stable episome. cccDNA is responsible for the persistence of the virus in the host cells and serves as the template for all viral mRNAs [53]. The main transcription product, HBx protein, acts as an activator for various host cellular genes crucial for both HBV replication and hepatocarcinogenesis by regulating DNA repair mechanisms and cell growth. In fact, studies analyzing whole-genome sequencing in HBV-related liver cancer have identified heightened levels of copy number variations at specific gene locations (breakpoint) where HBV integrates into the genome. This finding suggests that HBV integration likely triggers chromosomal instability, further implicating its role in carcinogenesis [54]. HBx protein contributes to the hepatocellular cycle dysregulation through several mechanisms, facilitated by its interaction with many intracellular pathways that modulate cell proliferation, cell death, transcription, and DNA repair. Specifically, it interacts with cAMP response element-binding (CREB) protein/P300 (CBP/P300), directly influencing CREB-dependent transcription. It impacts transcription by involving cellular signaling pathways such as Ras/Raf, mitogen-activated protein kinase (MAPK), and Janus kinase (JAK)—signal transducer of activators of transcription (STAT). Finally, HBx protein also influences proteasomes, mitochondrial proteins, p53, and DDB1, leading to its apoptotic effects [53,55]. The resulting genetic instability forms the basis for the neoplastic transformation of the host cell. The multifunctional nature of HBx causes the alteration in several fundamental cellular mechanisms and induces the proliferation of tumorigenic traits capable of inducing HCC [56]. Moreover, HBx promotes HCC invasion and metastasis both in vitro and in vivo with its oncogenic activity, thereby suggesting that HBx could be used as a novel target for HCC therapy [57].

Another viral protein involved in the carcinogenetic process is HBV core protein (HBc), which is the major capsid protein of the virus. Several studies showed that HBc acts as an important mediator of hepatocarcinogenesis through several mechanisms including the promotion of apoptosis resistance and the repression of proapoptotic factors [58]. Moreover, the expression of HBc promotes the proliferation of hepatoma cells in vitro through the activation of the Src/PI3K/Akt pathway [59]. Finally, HBc seems to act synergistically with HBx, repressing the promoter activity of the p53 gene and inducing liver cancer.

In conclusion, the oncogenic role of HBV is due to the coexistence of direct and indirect mechanisms. The incidence of HBV-related HCC significantly varies depending on the infection status and the stage of liver disease. DNA integration promoting genome instability is the mainstay factor in patients without cirrhosis that could lead to carcinogenesis process. This underscores the multifactorial nature of HBV-associated carcinogenesis and highlights the importance of improving future research in this context. Therefore, a comprehensive understanding of these diverse mechanisms and host-related variables is essential for effective management and prevention strategies targeting HBV-related HCC.

### 3.2. Incidence and Risk Factors of HCC in Patients with HBV

The virus-dependent biological factors associated with a more aggressive oncogenesis are HBe antigen seropositivity, high viral load, and genotype C [60]. The carcinogenic process linked to genotype C could be associated with basal core promoter mutations [55]. Additionally, patients with genotype C frequently show higher HBe antigen levels, potentially explaining the more aggressive disease course [61]. In 2006, Chen at al. demonstrated the crucial association between the viral load and the HBV-linked hepatocarcinogenesis. The HCC risk was associated with high HBV DNA levels in serum, and higher the level, the stronger the association with HCC, even in patients negative for the HBe antigen [62].

Regarding clinical and epidemiological data, chronic HBV infection represents the major etiological risk factor for HCC development worldwide, with approximately half of the patients with HBV-related HCC [63]. The 5-year expected cumulative incidence of cirrhosis in patients with untreated chronic hepatitis B (CHB) is 8–20%, while the annual HCC risk in patients with HBV-related cirrhosis is estimated to be 2–5% [60].

Although antiviral therapies, mostly nucleos(t)ide analogues (NAs)-based therapies, have profoundly changed the natural history of HBV infection, virus elimination remains challenging. To date, the most effective public health measure has been the implementation of vaccination [64]. Indeed, vaccination prevents virus infection and, as consequence, genome integration, which is the key aspect of oncogenic promotion.

### 3.3. HCC Risk in Untreated and Treated Patients

The HCC risk in untreated patients with HBV depends on the HBV status: inactive carrier (comparable to chronic HBe antigen-negative infection, according to the most recent classification) has an incidence rate of 0.05 (95% CI: 0.03–0.08) per 100 person-years, patients with chronic hepatitis B have an incidence rate of 0.42 (95% CI: 0.27–0.56) per 100 person-years, and patients with compensated cirrhosis exhibit an incidence rate of 2.97 (95% CI: 2.35–3.59) per 100 persons-years, nearly 60-fold higher than patients with a chronic infection (in Europe, the subtotal incidence rates were significantly lower: 0.03, 0.12, and 2.03, respectively) [65]. These findings underscore that the HCC risk is strictly linked to liver status and to advanced fibrosis or cirrhosis. In a prospective cohort of nearly 2000 on-therapy patients with CHB (with NAs: entecavir, ETV, and tenofovir disoproxil fumarate, TDF), for 5 to 12 years after NAs were started, HCC developed in 1.2% of patients without cirrhosis at baseline and in 5.75% of patients with cirrhosis at baseline. The HCC risk after the first 5 years of antiviral therapy depends on age, baseline cirrhosis status, and liver stiffness measurement at 5 years [66]. NA-based therapies aim to achieve long-term suppression of viral load, HBeAg loss, and a seroconversion from HB antigens to anti-HB antigens in order to minimize liver disease progression, liver-related events, and the risk of HCC incidence. NA-based therapy has shown superiority over interferon (INF)-based therapy in reducing the incidence of HCC compared to controls. In fact, the RR obtained from INF-based therapies ranged from 0.50 (95% CI: 0.05–0.94) to 0.66 (95% CI: 0.48–0.89), while that obtained from NAs ranged from 0.22 (95% CI: 0.10–0.50) to 0.55 (95% CI: 0.31–0.99), *p* < 0.001 [67]. TDF treatment in a retrospective HBV-related cirrhosis cohort was independently associated with a lower HCC risk (aHR: 0.46) [68]. Liver status also influences the HCC risk due to antiviral therapy: the 5-year HCC cumulative incidence was 0.5–6.9% in on-therapy patients without cirrhosis, 4.5–21.6% with compensated cirrhosis, and 36.3–46.5% with decompensated cirrhosis, considering the significantly decreased annual incidence rate within 4 versus 4–8 years (0.2% versus 0% in the low-risk, 1.1% versus 0.2% in the intermediate-risk, and 4.6% versus 1% in the high-risk groups, respectively) [69].

Table 2 presents the most recent and accurate studies that evaluated HCC incidence in patients with HBV.

In the literature, the fact that an antiviral drug is most effective in reducing the HCC risk is still debated. Firstly, Choi et al. [70] showed a lower HCC risk in those who received TDF compared to ETV-treated cases. Several studies and meta-analyses, mostly conducted in Eastern countries, seem to support the superiority of TDF over ETV [71]. Among the most recent and detailed studies, CHB patients receiving TDF had a significantly lower HCC risk (adjusted HR: 0.77; 95% CI: 0.61–0.98) than those receiving ETV, particularly in patients older than 50 years (HR 0.76, 95% CI 0.58–1.00), males (HR 0.74, 95% CI: 0.58–0.96), and individual who were HBeAg positive (HR 0.69, 95% CI: 0.49–0.97) [72]. However, these data were not confirmed by the largest prospective study conducted in France [73] on 1800 patients with CHB (986 patients treated with TDF and 814 patients treated with ETV). They concluded that the risk of liver-related events or death did not differ between patients treated with TDF and ETV, based on a 4-year median follow-up. Therefore, it is not possible to infer the superiority of TDF over ETV with absolute certainty, and prospective studies on homogeneous populations with CHB are needed, also considering longer follow-up periods.
cancers-16-01505-t002_Table 2Table 2Studies evaluating HCC incidence in patients with HBV.Author,JournalCountryMethodologyPatient CharacteristicFindingsPapatheodoridis GV,J Hepatol 2020 [66]EuropeRetrospective370 (26.9%) patients with cirrhosisAll treated patientsHCC occurrence: 1.2% of patients with chronic hepatitis vs. 5.75% of patients with cirrhosisLiu K,APT 2019 [68]ChinaRetrospective797 patients treated with TDF vs. 291 untreated patients53.7% patients with cirrhosis5-year cumulative probability of HCC: 14.9% in untreated patients vs. 9.8% in patients treated with TDFChoi J,Jama Oncol 2019 [70]KoreaRetrospective11,464 patients treated with ETV and 12,692 patients treated with TDFCirrhosis: 26.1% in ETV vs. 27.5% in TDFAnnual incidence rate of HCC: 1.06 per 100 p/y in ETV vs. 0.64 in TDF groupsPol S,APT 2021 [73]FranceRetrospective814 patients treated with ETV and 986 patients treated with TDFCirrhosis: 9% in both groupsHCC incidence rate: 1.6 per 100 p/y in ETV vs. 1.8 per 100 p/y in TDF groups (not a statistically significant difference)Abbreviations: HCC: hepatocellular carcinoma; TDF: tenofovir disoproxil fumarate; ETV: entecavir; and p/y: person-year.

### 3.4. Occult Hepatitis B Virus Infection (OBI) and HCC Risk

A not fully explored and understood entity is Occult Hepatitis B Infection (OBI), in which replication-competent HBV DNA should be present in the liver, and patients usually exhibit HBsAg negativity, with or without serum viral load detection [74]. Its clinical relevance is attributed to the integration of life-long DNA into the host cells [75]. An examination of liver tissues from patients with HCC without HBsAgs and with anti-HBc antigens showed that most of these patients had a significantly higher prevalence of HBV DNA compared to tissue from patients without HCC [76]. Moreover, in a retrospective HBsAg-negative cohort [77], after male sex, the HBV-DNA positivity was the second strongest predictors for carcinogenesis (HR: 8.25, 95% CI: 2.01–33.93). One of the largest studies in this field was conducted on 609,299 patients undergoing hepatitis B serology examination, with a 9-year median follow-up, aimed to investigate liver-related and liver cancer mortality [78]. As expected, patients with a current HBV infection had the worst prognosis (a liver-related mortality rate of 129.6/10^5^ person years). However, patients with isolated anti-HBc positivity exhibited higher mortality compared to patients with anti-HBc positivity associated with anti-HBs positivity: the liver-related mortality rate was 22.5 vs. 7 per 10^5^ person-years, and the liver cancer mortality rate was 16.8 vs. 4.0 per 10^5^ person-years. A definitive direct association between OBI and the HCC risk has not been established, and it could be influenced by virus replication. Defining the clinical significance of OBI, particularly its role in hepatocarcinogenesis and in accelerating progression to cirrhosis in patients with other identifiable causes of liver disease, as well as those without identifiable causes, will be listed in our future research.

## 4. Hepatitis D Virus

### 4.1. Mechanisms of Hepatocarcinogenesis

The exact mechanisms of HDV virus-related oncogenesis are not completely understood, even if it is well known that chronic hepatitis D is associated with an increased risk of developing HCC compared to an HBV mono-infection [79]. Indeed, it is unknown if HCC is the result of a cumulative effect of both HBV and HDV, an effect of the underlying cirrhosis, or a direct oncogenic effect of HDV [80]. A direct oncogenic effect of HDV is unlikely because the virus does not integrate into the human genome. The increased incidence of HCC in patients with HDV compared to those with a HBV mono-infection is probably due to more severe inflammation, oxidative stress, and DNA damage, leading to an accelerated liver fibrosis progression to cirrhosis [81]. However, recently, it has been suggested that the molecular mechanism that supports the oncogenic potential of HDV is different from that triggered by HBV. Several molecular pathways could potentially be implicated in HDV hepatocarcinogenesis. Initial studies suggest that HDV may modify the fibrosis-related signaling pathway of tumor growth factor beta (TGF-ꞵ), with a role in accelerating co-infected cells and as a regulator in fibrosis and hepatocarcinogenesis. The principal mechanisms by which HDV promotes HCC development are through the modifications in innate immune responses, the stimulation of adaptive immune responses, the epigenetic alterations, and/or the production of reactive oxygen species (ROS) [37]. Large delta antigen (L-HDAg) could promote hepatocarcinogenesis by the activation of TGF-β pathway, via the Smad3 protein. Moreover, L-HDAg can also induce NF-κB activation and ROS production, stimulating tumor necrosis factor-α (TNF-α) [82,83].

Recently, Diaz et al. [84] provided the first molecular signature of HDV-related liver tumor. The genetic instability has a crucial role in the HDV hepatocarcinogenesis as demonstrated by transcriptomic gene expression analysis performed by comparing patients with HDV-related HCC to patients with HDV cirrhosis without HCC, by evaluating tumor and non-tumor liver tissues. This large network of genes identified are functionally related to DNA repair, cell cycle, mitotic apparatus, and cell division. Despite the dependence of HDV on HBV, these findings suggest that the mechanisms related to HDV carcinogenesis are different from those related to HBV.

In conclusion, the answer to the question whether the delta virus is an oncogenic virus seems to be more possible than in the past but is still under studied. Therefore, well-designed prospective cohort studies comparing patients with HDV/HBV in comparison to those with a HBV mono-infection will be key in understanding the oncogenic properties of the virus.

### 4.2. HCC Risk in Patients with HBV/HDV

Studies aimed to evaluate the association between HDV and HCC are few and are frequently retrospective, with suboptimal designs. Thus, the impact of HDV infection on HCC development is controversial. Stockdale et al. estimated that, globally, between 15% and 20% of cirrhotic or HCC cases among people with HBV are attributable to HDV infection, indicating that it may be an important contributor to severe liver disease, accelerating disease progression [85]. HDV may have an even greater attributable fraction of HBV-associated mortality from cirrhosis and HCC. Particularly, patients with HCC had a higher anti-HDV prevalence compared to patients without HCC, considering a homogeneous geographic distribution (pooled ORs: 4.8). Among people positive for HBs antigen with evidence of liver disease progression, HDV accounted for 20% of HCC (95% CI: 8%–33%) [78]. In a cohort of 188 Italians (most of them had cirrhosis at presentation and were observed for a mean period of 7.8 years), the median probability of survival free of major events, either decompensation or HCC, from the initial diagnosis, was 28 years [86]. In two longitudinal studies, the annual incidence rates for HCC were 1% to 2.8% [86,87]. Additionally, Fattovich et al. showed a three times higher HCC risk in patients with HDV-related cirrhosis than in patients with HBV-related cirrhosis [88], indicating that HDV is as a major driver for HCC development in patients positive for HBs antigen.

A recent meta-analysis showed that an HBV/HDV co-infection was more frequently associated with HCC (9.7%) rather than an HBV mono-infection (5.1%) [79]. Similarly, Alfaiate et al., in their study, obtained the same results, confirming a pooled odds ratio of 2.77 for HCC development in patients with HDV compared with those with HBV mono-infection [89].

A recent Italian study evaluating the clinical outcome of the long-term treatment with either TDF or ETV in a cohort of patients with a HDV co-infection showed that a co-infection creates a significantly greater risk of liver-related complications compared to a mono-infection, and the incidence of HCC was almost thrice in the HDV cohort (3.12 vs. 1.12; *p* = 0.02), with an estimated annual rate of HCC development of 7.5% in patients with HDV and 2.5% in controls (*p* = 0.01) [90].

The recent approval of bulevirtide (BLV) as an entry inhibitor for the specific treatment of HDV infection could probably have a great impact on liver-related events, particularly on hepatocarcinogenesis. Until now, the literature has suggested that a BLV-based treatment leads to continued RNA decline and the improvements in biochemical disease activity. Future studies with longer follow-ups are needed to evaluate the effects of BLV on the occurrence of HCC.

In conclusion, patients with co-infection HBV/HDV had a significantly higher risk of HCC than those with HBV mono-infection, and it requires a tailored follow-up.

## 5. Conclusions

HCC occurrence is a complicated process affected by various factors. Viral hepatitis represents an important predisposing factor for liver carcinogenesis towards chronic inflammation, epithelial-to-mesenchymal transition, and overt fibrosis and cirrhosis. Chronic viral infection and immune-mediated damage changes the liver microenvironment, contributing to the strongest risk factors for HCC development. It is also mandatory to consider all the potential hepatocarcinogenetic pathways in this field, including the direct mechanisms, because a non-negligible percentage of HCC occurs in patients without a cirrhotic substrate. Current treatment options are the mainstay in the primary and secondary prevention of HCC, even if they reduce the risk without eliminating it completely. Indeed, while a portion of patients with HCV after SVR may be subjected to a minor HCC risk, all patients with HBV, even with sustained virological suppression, must undergo HCC surveillance, especially those with an HBV/HDV co-infection. Future research may focus on stratifying better high-risk patients and, of course, on understanding the potential impact of bulevirtide on hepatocarcinogenesis. On the other hand, it is well known that not all patients with chronic viral infection develop HCC, suggesting that additional factors are involved, including other factors (such as the emerging metabolic syndrome) and also the host responses. Future studies aimed at comprehending the influence of these viruses on the host immuno-response may provide new perspectives on HCC occurrence as well as new therapeutic targets, to limit or prevent the liver disease progression.

## Figures and Tables

**Figure 1 cancers-16-01505-f001:**
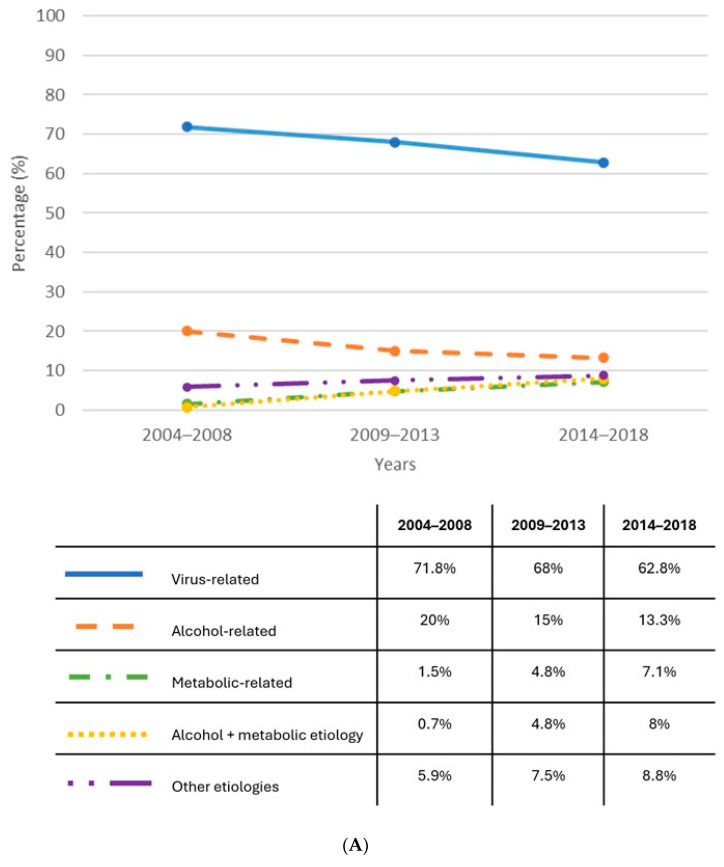
Italian temporal trends from 2004 to 2018 of the proportion (%) of all etiologies for HCC: (**A**) description of temporal trends of all etiologies for HCC, also including non-viral causes and (**B**) a specific focus on the proportion of virus-related HCC [6].

**Table 1 cancers-16-01505-t001:** Studies evaluating HCC incidence in patients with HCV.

Author, Journal	Country	Methodology	Patients with Cirrhosis	Findings
Calvaruso V, Gastroenterology 2018 [46]	Italy	Prospective	2249 patients (100%)	HCC occurrence: 3% in SVR vs. 12.8% in non-SVR (*p* < 0.001) HCC overall cumulative rate at 1 year: 2.9% in SVR vs. 8% in non-SVR
Kanwal F, Gastroenterology 2017 [47]	USA	Retrospective	8766 patients (39%)	HCC incidence: 0.9 per 100 p/y in SVR vs. 3.45 per 100 p/y in non-SVR
El-Serag HB, Hepatology 2016 [48]	USA	Retrospective	1548 patients (14.4%)	HCC incidence: 0.93 per 100 p/y in SVR vs. 3.27 per 100 p/y in non-SVR
Morisco F, Cancers 2021 [49]	Italy	Prospective	706 patients (100%)	Liver-related events: 8.9% in SVR vs. 26.3% in non-SVR HCC incidence in SVR: 1.6 per 100 p/y
Kondili L, DLD 2023 [50]	Italy	Retrospective	2064 patients (100%)	HCC incidence in SVR: 2.45 per 100 p/y
Kanwal F, Hepatology 2020 [51]	USA	Retrospective	6938 patients (38.4%)	HCC incidence in SVR: 1.23 per 100 p/y

Abbreviations: HCC, hepatocellular carcinoma; SVR, sustained virological response; and p/y, person-year.

## Data Availability

Not applicable.

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
