# Peer review of "The Role of Hepatitis Viruses as Drivers of Hepatocancerogenesis"

_cancers, 2024, doi:10.3390/cancers16081505_

Round 1

Reviewer 1 Report

Comments and Suggestions for Authors

It is an interesting manuscript about “The role of hepatitis viruses as drivers of hepatocancerogenesis ”.

My concern is determined in the following points.

HCC occurrence is a complicated process affected by various factors. Viral hepatitis represents an important predisposing factor for liver carcinogenesis towards chronic inflammation, epithelial to mesenchymal transition, and overt fibrosis and cirrhosis. Chronic viral infection and immune-mediated damages changes the liver microenvironment, contributing to the strongest risk factors for HCC development. Current treatment options are the mainstay in primary and secondary prevention for HCC, even if they reduce the risk without eliminate it at all. On the other hand, it is well known that not all patients with chronic viral infection develop HCC, suggesting that additional factors are involved, including other factors (such as the emerging metabolic syndrome) and as well  known, the host responses. Future studies aimed at comprehending the influence of these viruses on the host immuno-response may provide new prospectives in HCC occurrence as well as new therapeutic targets to limit or prevent the liver disease progression.

* HBV infection is associated with the risk of developing HCC with or without an underlying liver cirrhosis, due to various direct and indirect mechanisms promoting hepatocarcinogenesis. The molecular profile of HBV-HCC is extensively and continuously under study, and it is the result of altered molecular pathways, which modify the microenvironment and lead to DNA damage. HBV produces the protein HBx, which has a central role in the oncogenetic process. Furthermore, the molecular profile of HBV-HCC was recently discerned from that of HDV-HCC, despite the obligatory dependence of HDV on HBV. Proper management of the underlying HBV-related liver disease is fundamental, including HCC surveillance, viral suppression, and application of adequate predictive models. When HBV-HCC occurs, liver function and HCC characteristics guide the physician among treatment strategies but always considering the viral etiology in the treatment choice.

*The pathogenesis of HBV-mediated hepatocarcinogenesis is unclear. Evidence currently available suggests that the HBV core protein (HBc) plays a potential role in the development of HCC, such as the HBV X protein. The core protein, which is the structural component of the viral nucleocapsid, contributes to almost every stage of the HBV life cycle and occupies diverse roles in HBV replication and pathogenesis. Recent studies have shown that HBc was able to disrupt various pathways involved in liver carcinogenesis: the signaling pathways implicated in migration and proliferation of hepatoma cells, apoptosis pathways, and cell metabolic pathways inducing the development of HCC; and the immune system, through the expression and production of proinflammatory cytokines. In addition, HBc can modulate normal functions of hepatocytes through disrupting human host gene expression by binding to promoter regions. This HBV protein also promotes HCC metastasis through epigenetic alterations, such as micro-RNA. 

*HBV may induce HCC through the induction of chronic liver inflammation, which can cause oxidative stress and DNA damage. However, many studies also indicated that HBV could induce HCC via the alteration of hepatocellular physiology that may involve genetic and epigenetic changes of the host DNA, the alteration of cellular signaling pathways, and the inhibition of DNA repair mechanisms. This alteration of cellular physiology can lead to the accumulation of DNA damages and the promotion of cell cycles and predispose hepatocytes to oncogenic transformation.

*Chronic infection with hepatitis B virus (HBV), hepatitis delta virus (HDV), and hepatitis C virus (HCV) are the greatest etiological risk factors for HCC. Due to the significant role of chronic viral infection in HCC development, it is important to investigate direct (viral associated) and indirect (immune-associated) mechanisms involved in the pathogenesis of HCC. Common mechanisms used by HBV, HCV, and HDV that drive hepatocarcinogenesis include persistent liver inflammation with an impaired antiviral immune response, immune and viral protein-mediated oxidative stress, and deregulation of cellular signaling pathways by viral proteins. DNA integration to promote genome instability is a feature of HBV infection, and metabolic reprogramming leading to steatosis is driven by HCV infection. 

Above mentioned should be referred to.

Author Response

HCC occurrence is a complicated process affected by various factors. Viral hepatitis represents an important predisposing factor for liver carcinogenesis towards chronic inflammation, epithelial to mesenchymal transition, and overt fibrosis and cirrhosis. Chronic viral infection and immune-mediated damages changes the liver microenvironment, contributing to the strongest risk factors for HCC development. Current treatment options are the mainstay in primary and secondary prevention for HCC, even if they reduce the risk without eliminate it at all. On the other hand, it is well known that not all patients with chronic viral infection develop HCC, suggesting that additional factors are involved, including other factors (such as the emerging metabolic syndrome) and as well known, the host responses. Future studies aimed at comprehending the influence of these viruses on the host immuno-response may provide new prospectives in HCC occurrence as well as new therapeutic targets to limit or prevent the liver disease progression.

* HBV infection is associated with the risk of developing HCC with or without an underlying liver cirrhosis, due to various direct and indirect mechanisms promoting hepatocarcinogenesis. The molecular profile of HBV-HCC is extensively and continuously under study, and it is the result of altered molecular pathways, which modify the microenvironment and lead to DNA damage. HBV produces the protein HBx, which has a central role in the oncogenetic process. Furthermore, the molecular profile of HBV-HCC was recently discerned from that of HDV-HCC, despite the obligatory dependence of HDV on HBV. Proper management of the underlying HBV-related liver disease is fundamental, including HCC surveillance, viral suppression, and application of adequate predictive models. When HBV-HCC occurs, liver function and HCC characteristics guide the physician among treatment strategies but always considering the viral etiology in the treatment choice.

            We thank the reviewer for his comment. We agreed with the central role of HBx protein in the liver carcinogenesis and in the HCC invasion. Therefore, we added in the text also a comment on his potential role as target for HCC therapy.

*The pathogenesis of HBV-mediated hepatocarcinogenesis is unclear. Evidence currently available suggests that the HBV core protein (HBc) plays a potential role in the development of HCC, such as the HBV X protein. The core protein, which is the structural component of the viral nucleocapsid, contributes to almost every stage of the HBV life cycle and occupies diverse roles in HBV replication and pathogenesis. Recent studies have shown that HBc was able to disrupt various pathways involved in liver carcinogenesis: the signaling pathways implicated in migration and proliferation of hepatoma cells, apoptosis pathways, and cell metabolic pathways inducing the development of HCC; and the immune system, through the expression and production of proinflammatory cytokines. In addition, HBc can modulate normal functions of hepatocytes through disrupting human host gene expression by binding to promoter regions. This HBV protein also promotes HCC metastasis through epigenetic alterations, such as micro-RNA.

            We thank the reviewer for his suggestion and we added in the text a paragraph on the HBc role in liver carcinogenesis.

*HBV may induce HCC through the induction of chronic liver inflammation, which can cause oxidative stress and DNA damage. However, many studies also indicated that HBV could induce HCC via the alteration of hepatocellular physiology that may involve genetic and epigenetic changes of the host DNA, the alteration of cellular signaling pathways, and the inhibition of DNA repair mechanisms. This alteration of cellular physiology can lead to the accumulation of DNA damages and the promotion of cell cycles and predispose hepatocytes to oncogenic transformation.

            Thanks for the comment. We better clarify this aspect in the text.

*Chronic infection with hepatitis B virus (HBV), hepatitis delta virus (HDV), and hepatitis C virus (HCV) are the greatest etiological risk factors for HCC. Due to the significant role of chronic viral infection in HCC development, it is important to investigate direct (viral associated) and indirect (immune-associated) mechanisms involved in the pathogenesis of HCC. Common mechanisms used by HBV, HCV, and HDV that drive hepatocarcinogenesis include persistent liver inflammation with an impaired antiviral immune response, immune and viral protein-mediated oxidative stress, and deregulation of cellular signaling pathways by viral proteins. DNA integration to promote genome instability is a feature of HBV infection, and metabolic reprogramming leading to steatosis is driven by HCV infection.

            Thanks for the comment. We underlined through the review that hepatitis virus B, C and D promote liver cancer through direct and indirect mechanisms. Particularly, all these viruses caused liver fibrosis and cirrhosis through a persistent liver inflammatory process induced by cytokines and proliferative pathways directly activated by viruses.

Reviewer 2 Report

Comments and Suggestions for Authors

The current review addresses while metabolic liver disease is increasingly causing HCC, viral hepatitis B and C remain the major culprits. This review explores how these viruses directly promote cancer, especially during active infections, and how antiviral treatments can impact HCC risk.

Generally, this review effectively addresses the key role of hepatitis B and C viruses in driving liver cancer. The topic remains highly relevant due to the ongoing global burden of viral hepatitis. It provides a valuable update on viral mechanisms and the impact of antiviral therapy on HCC risk, complementing existing knowledge. Including studies on the interaction between viral hepatitis and metabolic risk factors could strengthen future research. The conclusions align with the presented evidence and the main question is comprehensively addressed. The references appear appropriate and up-to-date.

Minor concerns:

The authors' statement seems to be a missed opportunity regarding the longitudinal course of viral eradication. Please try concisely rephrasing the review as appropriate and feasible by addressing how factors like necroinflammation, fibrosis, stiffness, and portal hypertension change over time after successful treatment compared to pre-treatment with active infection.

Clinicians worldwide are encountering a growing number of patients who have eradicated their viral infection. For these healthcare providers managing a large body of post-treatment patients, a statement lacking temporal considerations reflecting the viral eradication process is not sufficiently interesting, informative, or practical.

Comments on the Quality of English Language

Minor editing of English language required

Author Response

The current review addresses while metabolic liver disease is increasingly causing HCC, viral hepatitis B and C remain the major culprits. This review explores how these viruses directly promote cancer, especially during active infections, and how antiviral treatments can impact HCC risk.

Generally, this review effectively addresses the key role of hepatitis B and C viruses in driving liver cancer. The topic remains highly relevant due to the ongoing global burden of viral hepatitis. It provides a valuable update on viral mechanisms and the impact of antiviral therapy on HCC risk, complementing existing knowledge. Including studies on the interaction between viral hepatitis and metabolic risk factors could strengthen future research. The conclusions align with the presented evidence and the main question is comprehensively addressed. The references appear appropriate and up-to-date.

Minor concerns:

The authors' statement seems to be a missed opportunity regarding the longitudinal course of viral eradication. Please try concisely rephrasing the review as appropriate and feasible by addressing how factors like necroinflammation, fibrosis, stiffness, and portal hypertension change over time after successful treatment compared to pre-treatment with active infection.

            We thank the reviewer for his comment. The aim of our review is to explain the role of hepatitis viruses in liver carcinogenesis and how the HCC risk is modified, but not eliminated, by viral eradication. Indeed, the majority of the studies showed that the HBV viral suppression by NUC therapy and the HCV eradication by DAAs reduce the liver inflammation due to viral replication and thereby the HCC risk. In our opinion, the addition, in the text, of a paragraph addressing changes in liver stiffness or portal hypertension after antiviral therapy is not in line with the aims of this review.

Clinicians worldwide are encountering a growing number of patients who have eradicated their viral infection. For these healthcare providers managing a large body of post-treatment patients, a statement lacking temporal considerations reflecting the viral eradication process is not sufficiently interesting, informative, or practical.

            Thanks for the comment. We agreed with your suggestion and we cited in the text a very recent paper analyzing the HCC risk, in cirrhotic patients, over time after SVR. The authors concluded that in the first 6 years after SVR, the HCC risk progressively declined, but remain stable after 6 years. Therefore, also many years after SVR, cirrhotic patients should continue the HCC surveillance. In our opinion, in the setting of a review, this is a crucial information for clinicians.

Minor editing of English language required.

            Done.

Reviewer 3 Report

Comments and Suggestions for Authors

I have gone over this review, defining the continued importance of hepatitis viruses in the causation of HCC amongst the increasing proportion of MASLD CLD and HCC. The authors have defined the current scenario about the statistics of data and stressed that hepatitis viruses continue to be the cause of HCC in over two-thirds of patients. Next, the authors have defined the pathogenesis of causation of HCC in chronic HCV, HBV, and HDV infection. I did enjoy reading this review. It looks informative to those who practice hepatology. 

Author Response

I have gone over this review, defining the continued importance of hepatitis viruses in the causation of HCC amongst the increasing proportion of MASLD CLD and HCC. The authors have defined the current scenario about the statistics of data and stressed that hepatitis viruses continue to be the cause of HCC in over two-thirds of patients. Next, the authors have defined the pathogenesis of causation of HCC in chronic HCV, HBV, and HDV infection. I did enjoy reading this review. It looks informative to those who practice hepatology.

We thank the reviewer for his comment.

Reviewer 4 Report

Comments and Suggestions for Authors

This manuscript deals with the review of the role of hepatitis viruses as drivers of hepatocancerogenesis. The manuscript is well-written and well-organized. Some information was very interesting for me and could be interesting for other readers. However, some suggestions listed below may improve the manuscript. 

1) More references should be cited in the Introduction. For example, I believe that after the sentence " Even though, it represents the third cause of 35 cancer-related mortality worldwide (8.3% of all sites)" more references could be added, not only reference 2. 

2) Maybe results presented in Figs,1a,b could be presented in one Figure. 

3) The paragraphs above "Viral pathogenesis" and "Incidence and risk factors of HCC in HBV patients", respectively contain only one sentence. They should be rewritten by adding more sentences. 

4) The paragraph above "Table 1" contains only one sentence. It should be rewritten by adding more sentences. 

5) The abstract is too long, while the Conclusion is too short. So, the Conclusion should be rewritten. 

Author Response

This manuscript deals with the review of the role of hepatitis viruses as drivers of hepatocancerogenesis. The manuscript is well-written and well-organized. Some information was very interesting for me and could be interesting for other readers. However, some suggestions listed below may improve the manuscript.

1) More references should be cited in the Introduction. For example, I believe that after the sentence " Even though, it represents the third cause of cancer-related mortality worldwide (8.3% of all sites)" more references could be added, not only reference 2.

Thanks for the suggestion. We modified the text as requested.

2) Maybe results presented in Figs,1a,b could be presented in one Figure.

Thanks for the comment. We previous tried to present only one Figure with a. and b. integration, however this hypothetic Figure results slightly chaotic. In this form we successfully show all factors (Fig 1a.) and specifically viruses (Fig 1b.) etiologies.

3) The paragraphs above "Viral pathogenesis" and "Incidence and risk factors of HCC in HBV patients", respectively contain only one sentence. They should be rewritten by adding more sentences.

Thanks, we have rewritten the paragraphs.

4) The paragraph above "Table 1" contains only one sentence. It should be rewritten by adding more sentences.

Thanks, we have rewritten the paragraph.

5) The abstract is too long, while the Conclusion is too short. So, the Conclusion should be rewritten.

Thanks, we have rewritten with more accuracy conclusions.